# From Development to Regeneration: Insights into Flight Muscle Adaptations from Bat Muscle Cell Lines

**DOI:** 10.3390/cells14151190

**Published:** 2025-08-01

**Authors:** Fengyan Deng, Valentina Peña, Pedro Morales-Sosa, Andrea Bernal-Rivera, Bowen Yang, Shengping Huang, Sonia Ghosh, Maria Katt, Luciana Andrea Castellano, Lucinda Maddera, Zulin Yu, Nicolas Rohner, Chongbei Zhao, Jasmin Camacho

**Affiliations:** 1Stowers Institute for Medical Research, Kansas City, MO 64110, USA; fdeng@stowers.org (F.D.);; 2Department of Ecology and Evolutionary Biology, University of California, Irvine, CA 92697, USA; 3Department of Cell Biology and Physiology, University of Kansas Medical Center, Kansas City, KS 66160, USA; 4Department of Biology, University of Washington, Seattle, WA 98195, USA; 5Institute for Integrative Cell Biology and Physiology, University of Münster, 48149 Münster, Germany

**Keywords:** bat, flight muscle biology, myoblast cell line, myotube, *hTERT*, *CDK4*, proliferation and differentiation, regeneration

## Abstract

Skeletal muscle regeneration depends on muscle stem cells, which give rise to myoblasts that drive muscle growth, repair, and maintenance. In bats—the only mammals capable of powered flight—these processes must also sustain contractile performance under extreme mechanical and metabolic stress. However, the cellular and molecular mechanisms underlying bat muscle physiology remain largely unknown. To enable mechanistic investigation of these traits, we established the first myoblast cell lines from the pectoralis muscle of *Pteronotus mesoamericanus*, a highly maneuverable aerial insectivore. Using both spontaneous immortalization and exogenous *hTERT/CDK4* gene overexpression, we generated two stable cell lines that retain proliferative capacity and differentiate into contractile myotubes. These cells exhibit frequent spontaneous contractions, suggesting robust functional integrity at the neuromuscular junction. In parallel, we performed transcriptomic and metabolic profiling of native pectoralis tissue in the closely related *Pteronotus parnellii* to define molecular programs supporting muscle specialization. Gene expression analyses revealed enriched pathways for muscle metabolism, development, and regeneration, highlighting supporting roles in tissue maintenance and repair. Consistent with this profile, the flight muscle is triglyceride-rich, which serves as an important fuel source for energetically demanding processes, including muscle contraction and cellular recovery. Integration of transcriptomic and metabolic data identified three key metabolic modules—glucose utilization, lipid handling, and nutrient signaling—that likely coordinate ATP production and support metabolic flexibility. Together, these complementary tools and datasets provide the first in vitro platform for investigating bat muscle research, enabling direct exploration of muscle regeneration, metabolic resilience, and evolutionary physiology.

## 1. Introduction

Understanding how skeletal muscle regenerates and maintains function under stress is critical for addressing muscle loss, injury, and degenerative diseases. Myoblasts, the progenitor cells of skeletal muscle, play central roles in both muscle development and repair [1,2,3]. While most models in muscle biology rely on human and rodent cells, expanding to non-traditional species offers an opportunity to uncover lineage-specific adaptations of regeneration and functional resilience, especially in mammals with extreme physiological traits (Table 1).

The evolution of powered flight in bats represents one of the most dramatic physiological transformations in mammalian history. As the only mammals capable of flight, bats underwent rapid morphological and functional innovations [18,19]. A hallmark of this transition is the specialization of skeletal muscle—not only for force production, but for sustaining the intense and continuous energy output during flight [20]. Although the developmental changes underlying bat wing morphology (e.g., elongation of the digits, formation of the wing membrane, and specialization of flight muscles) have been partially characterized [21,22,23,24,25,26,27], the molecular and metabolic programs that support the evolution of flight, particularly those related to muscle physiology and energy balance, remain poorly understood [28,29,30].

Bats represent a diverse group of mammals known for their distinctive physiological traits, including exceptional longevity without age-related reproductive decline [31,32], low cancer incidence [33,34], dampened inflammatory responses [35,36,37,38], viral resistance [38], rapid wound healing [39], enhanced DNA repair [40], and the capacity to enter hibernation or torpor [41]. These traits suggest that bats possess robust mechanisms for tissue protection and long-term physiological resilience. This makes them a compelling model for studying muscle biology, particularly in the context of regeneration, mitochondrial oxidative stress, and aging [31,32]. Their muscles, in particular, must remain functional under intense workloads, recover efficiently from damage, and resist age-related decline [42]. Recognizing this potential, establishing in vitro bat muscle models represents a key step toward uncovering the molecular mechanisms underlying their exceptional stress-tolerant physiology [43,44,45,46,47,48].

To investigate the cellular mechanisms that underlie these adaptations, tractable in vitro systems are needed. Recent studies have successfully derived and metabolically characterized bat cell lines from diverse tissues, including fibroblasts derived from heart and lung [49] or non-lethally from wing tissue [45,47], pluripotent stem cells (iPSC) from fibroblasts [50], kidney cells [51,52], and lung cells [46]. A full overview of currently available bat cell models is summarized in [48], which highlights cell lines established for studying immune function and virology, underscoring the growing use of bat-derived models in cellular and molecular studies. However, no muscle-specific immortalized bat cell models have yet been available, limiting our ability to directly examine how flight-adapted muscle supports energy-intensive behaviors like sustained flight, regeneration, and crosstalk with immune signaling pathways [29,53]. Developing a bat muscle cell model is therefore a crucial next step for understanding how bats achieve muscle performance and resilience under extreme physiological demands.

Culturing primary myoblasts poses a major challenge due to their limited lifespan in vitro [54]. Mammalian myoblasts rapidly lose their stemness shortly after isolation and rarely maintain it beyond 10 passages because of cellular senescence [55,56]. Senescence is primarily driven by two key mechanisms [11,57]. First, progressive telomere shortening occurs with each cell division until they reach a critical length that activates the p53 pathway and leads to cell senescence [58,59,60,61]. Telomerase reverse transcriptase (TERT), the catalytic subunit of the telomerase complex, maintains telomere length and delays senescence. Its expression in primary cells has been shown to extend lifespan and preserve proliferative potential [62,63]. Second, senescence is mediated by the cyclin-dependent kinase inhibitor (CDKI), p16, which inhibits CDK4/6 kinases’ activity and causes G1 cell cycle arrest [64]. Overexpression of CDK4 can bypass this arrest, maintaining cell proliferation while preserving normal cell functions [7,8,65]. Together, co-expression of TERT and CDK4 has enabled the successful immortalization of myoblasts in several species [7,10,11,15,65,66,67]. Despite this progress in immortalizing myoblasts from multiple species, no existing models capture the extreme physiological performance and tissue resilience observed in bats (Table 1).

Although there have been advances in comparative genomics in bats, functional models for the mechanistic investigation of bat muscle biology remain absent. This gap limits our ability to study the cellular and molecular basis of muscle performance, such as contraction, regeneration, strength, and fatigue resistance. To address this, our study pursued two complementary goals aimed at understanding skeletal muscle adaptations in bats. First, we performed transcriptomic profiling of the pectoralis major muscle in *Pteronotus parnellii* (*P. parnellii*), a Neotropical aerial insectivore [68], before and after acute glucose stimulation (5.4 g/kg body weight), a physiological proxy for the induced metabolic activity of flight muscle (Figure 1). This approach enabled us to define in vivo gene expression programs associated with energy production, oxidative resilience, and contractile function. We integrated this data with systemic glucose dynamics via glucose tolerance testing (GTT) and biochemical quantification of tissue energy stores (triglycerides and glycogen) to characterize muscle energetics in vivo. In parallel, to functionally explore these pathways and create an experimental platform for bat muscle biology, we established and characterized the first bat myoblast cell lines from *Pteronotus mesoamericanus* (*P. mesoamericanus*), a closely related species that also exhibits high-speed, high-endurance flight [69,70]. One cell line arose through spontaneous immortalization, while the other was engineered via hTERT/CDK4 overexpression to overcome replicative senescence.

While the RNA-seq and cell line experiments were performed in different species, their close phylogenetic relationship and shared ecological traits justify an integrated framework. Both species belong to a subset of *Pteronotus* bats that have evolved Doppler-sensitive sonar, a specialized echolocation system requiring rapid neuromuscular coordination [71,72]. By combining energy-responsive gene expression profiling in tissue with the development of bat muscle cell lines, our study offers both a system-level view into the evolutionary pressures of flight and a tractable cellular model to dissect molecular mechanisms of metabolic adaptation and muscle regeneration. Together, these data provide a framework for uncovering the cellular and molecular basis of muscle function under extreme physiological stress.

## 2. Materials and Methods

### 2.1. Sample Collection

*P. mesoamericanus* was selected for in vitro modeling based on its genomic signatures of stress tolerance, longevity, and immune modulation [73]. A single adult male, *P. mesoamericanus* (BZ#3524), was collected in April 2024 at the Lamanai Archaeological Reserve, Orange Walk District, Belize. Animal capture, handling, and euthanasia followed the American Society of Mammalogists’ guidelines for the ethical use of wild mammals in research [74]. All procedures were approved by the Institutional Animal Care and Use Committee at the American Museum of Natural History (AMNHIACUC-20240130). The bat was captured using ground-level mist nets and transported in an individual cloth bag to the Lamanai Field Research Center. Euthanasia was performed using isoflurane inhalation (<1 mL applied to a cotton ball) followed by thoracic compression. This method induces rapid unconsciousness and humane death within 1–2 min due to the high respiratory rate of bats. The bat tissue was sent to J.C. by the American Museum of Natural History for cell line development.

The pectoralis major muscle of *P. mesoamericanus* bat was dissected immediately post-euthanasia and rinsed three times in chilled 1× PBS, followed by obtaining four 5 mm circular biopsies using a sterile biopsy punch (Integra™ 3335, Integra Lifesciences, Princeton, NJ, USA) in a sterile petri dish filled with chilled 1× PBS. Two biopsies were flash frozen in liquid nitrogen for downstream RNA sequencing. The remaining two were briefly rinsed in chilled biopsy media (2% FBS + 1 mM EDTA in 1× PBS), then quickly transferred into chilled myoblast isolation media (DMEM/F12 with 15 mM HEPES buffer plus 1× MyoCult^TM^ Expansion Supplement, STEMCELL Technologies, Vancouver, BC, Canada). Each biopsy was placed into a sterile microcentrifuge tube containing 2.0 mL of myoblast media with 0.1% Collagenase type I (Sigma (St. Louis, MO, USA), Cat#: SCR103) and minced with surgical scissors (Fine Science Tools (Foster City, CA, USA), Cat#: 91408-14) until no visible tissue clumps remained. Tubes were incubated for 30 min in a 37 °C water bath. After enzymatic digestion, the tissue was mechanically dissociated by gentle trituration using a P1000 pipette. Cell suspensions were passed through a 100 μm cell strainer into a 12-well culture plate. The filtered cells were transferred into sterile microcentrifuge tubes, centrifuged at 350× *g* for 5 min at ambient temperature (~27 °C), and the supernatant was discarded. Pelleted cells were resuspended in 2.0 mL of myoblast media with 10% DMSO and transferred to cryovials. Cryovials were placed into a Corning^®^ CoolCell^®^ (Corning, NY, USA) and cooled (−1 °C/min) on dry ice for 1.5 h before storage and shipment in vapor-phase liquid nitrogen. Samples were collected and exported under Belize Forest Department permits FD/WL/7/24 (56) and FD/WL/7/24 (57). Cells were maintained at −80 °C until primary cell derivation.

### 2.2. RNA Sequencing

#### 2.2.1. RNA Extraction

Skeletal muscle tissue was obtained from male *P. parnellii* bats (*n* = 13), a species formerly considered conspecific with *P. mesoamericanus* [68]. Although a distinct species, *P. parnellii* shares key ecological, physiological, and behavioral traits with *P. mesoamericanus*, including large pectoralis muscles for flight. These individuals had previously undergone glucose tolerance testing during field studies in April 2022 [75] and were collected with the Trinidad Forestry Division and exported under permit #734 in collaboration with Alexa Sadier, PhD. The non-lethal GTT study was approved by the Stowers Institute for Medical Research (IACUCLASF2024-170).

As the primary site of post-prandial glucose uptake and major driver of energy demand during flight, the pectoralis is a key tissue for uncovering metabolic adaptations. Approximately 10 mg of flash-frozen pectoral muscle was homogenized in 0.4 mL of lysis buffer/homogenization solution using a BeadBug 6 microtube homogenizer (Benchmark Scientific, Sayreville, NJ, USA) containing zirconium oxide beads, bead size 1.5 mm. Total RNA was extracted following the manufacturer’s protocol for the Maxwell^®^ RSC simplyRNA Tissue Kit (Promega (Madison, WI, USA), Cat# AS1340). RNA concentration and integrity were assessed prior to sequencing.

#### 2.2.2. RNA Sequencing and Read Processing

Total RNA was submitted for poly-A–selected, stranded RNA sequencing using the NEBNext^®^ Ultra™ II RNA Library Prep Kit (Illumina, San Diego, CA, USA). Libraries were sequenced on the AVITI-75 platform (Element Biosciences, San Diego, CA, USA) using paired-end reads. Ribo-depleted, poly-A–enriched libraries were sequenced with temporal resolution to assess differential gene expression over time. Raw sequencing reads were demultiplexed using Illumina bcl-convert v3.10.5, allowing up to one mismatch per index. Reads were aligned to the *P. mesoamericanus* genome (UCSC GCF_021234165.1) using the STAR aligner (v2.7.10b) with NCBI_2023 gene model annotations. Transcript abundance was quantified as TPM using RSEM v1.3.1. Initial quality control included Spearman correlation analysis to assess consistency across biological replicates. Sample correlations were used to identify outliers and validate timepoint groupings. All raw and processed RNA-seq data are available at the NCBI Gene Expression Omnibus (GEO) under GSEID: GSE302800.

#### 2.2.3. Differential Gene Expression Analysis

Orthologous genes were identified using annotations from the NCBI RefSeq database to enable comparison of conserved transcriptional responses to glucose stimulation and facilitate functional interpretation based on better-characterized gene annotations in other species. Downstream analysis was performed in R v4.3.1 using the edgeR package (v3.36.0). Genes with low expression were filtered out, reducing the dataset from 22,050 annotated genes to 12,406 expressed genes. Normalization of read counts and differential expression testing were carried out in edgeR, with DEGs defined as those with an adjusted *p*-value < 0.05 and absolute log_2_ fold change > 1. GO annotations were obtained from NCBI RefSeq and applied to differentially expressed genes (DEGs) within each treatment group. GO enrichment analysis was performed using ClusterProfiler v4.10.0, with enrichment based on DEGs passing thresholds of *p* < 0.05 and absolute fold change > 1.5. Enriched biological processes and cellular components were used to contextualize gene expression patterns related to metabolic adaptation.

#### 2.2.4. Functional Profiling of Metabolically Stimulated Differentially Expressed Genes

To identify pathways relevant to muscle physiology and flight metabolism, we analyzed differentially expressed genes (DEGs) from *Pteronotus* pectoralis muscle following glucose administration (30 and 60 min post-feeding) compared to fasting (0 min). We filtered for genes with a log_2_ fold change > 1.5 (Appendix A) and conducted gene set enrichment analysis using WebGestalt 2024 [76] for Over-Representation Analysis (ORA) against KEGG, Reactome, PANTHER, and OMIM databases; results were visualized as dot plots (Figure 1E). To examine functional organization among DEGs, we used g:Profiler to identify pathway affiliations for each gene [77], then clustered genes based on shared enrichment profiles using Jaccard similarity. This analysis grouped DEGs into four major functional modules: glucose utilization, lipid handling, metabolic signaling, and muscle endurance.

#### 2.2.5. Candidate Gene Expression

Expression values were normalized to the mean expression of a panel of 13 housekeeping genes (*Actb*, *Gusb*, *Hprt1*, *Ipo8*, *Ppia*, *Rpl13a*, *Rpl19*, *Rps13*, *Rps18*, *Rps23*, *Tbp*, *Tubb*, *Ywhag*), selected for their stability across all conditions (Appendix A). This panel established a baseline reference for constitutive transcriptional activity. A curated set of candidate genes (Appendix A) was selected based on their established roles in fiber-type specification, redox buffering, calcium handling, stress signaling, and muscle regeneration. Significant differences from the housekeeping mean were initially assessed using Wilcoxon signed-rank tests (Appendix A). To assess dynamic responses to muscle stimulation, Δlog_2_CPM values were calculated for each gene by subtracting baseline expression (WT0) from expression at WT30 and WT60. These changes were then centered relative to the mean change observed among housekeeping genes to identify candidate genes with shifts exceeding background variation. For each gene, a “z-score” was calculated using the following formula:Z = Δcandidate − μHK/σHK
where μ and σ represent the mean and standard deviation of Δlog_2_CPM among housekeeping genes. Z-scores and two-tailed *p*-values were computed based on the housekeeping gene Δ distribution, followed by false discovery rate correction (Benjamini-Hochberg method). Genes with FDR < 0.05 were considered significantly responsive (Appendix A).

### 2.3. Isolation and Purification of Bat Primary Myoblasts

Cryopreserved cell suspensions stored at −80 °C were thawed and plated in a 24-well plate containing myoblast growth medium 1 (GM1) (Table 2). Cell cultures were maintained in a humidified incubator with 5% CO_2_ at 37 °C until cell confluency reached ~75%.

Myoblasts were enriched using the pre-plating method, which separates cells based on differential adhesion properties to the culture surface [78,79]. Briefly, cells were seeded in myoblast growth medium on a gelatin-coated culture dish and incubated at 37 °C with 5% CO_2_ for 20 min. This pre-plating step was repeated two additional times during subsequent subcultures to increase myoblast purity. All cells were tested negative for mycoplasma contamination.

### 2.4. Cell Immortalization

Bat primary myoblasts were either spontaneously immortalized or immortalized through lentiviral overexpression of human telomerase reverse transcriptase (*hTERT*) and cyclin-dependent kinase 4 (*CDK4*) genes. For *hTERT/CDK4*-mediated immortalization, primary myoblasts were transduced using lentiviral particles encoding genes *hTERT* (GenTarget Inc. (San Diego, CA, USA), Cat #: LVP1131-RB-PBS) and *CDK4* (GenTarget Inc., Cat #: LVP1140-RB-PBS), each carrying a blasticidin selection marker. Transduction was performed using the spinoculation method [80]. Briefly, 50,000 bat primary myoblasts were seeded in one well of a 24-well plate. On the second day, primary myoblasts were spinoculated at 1000× *g* for 2 h with *hTERT* and *CDK4* lentivirus, with an MOI of 20, in the presence of 8 μg/mL polybrene (Tocris Bioscience (Bristol, UK), 7711), after which, the media was replaced with fresh myoblast media. Then, myoblasts were incubated at 37 °C with 5% CO_2_. Two days post-transduction, cells were selected in 4 µg/mL blasticidin (InvivoGen (San Diego, CA, USA), Cat #: ant-bl-1) for 6 days, at which point all the untransduced control cells had died (Appendix A). Both hTERT/CDK4-immortalized myoblasts (iBatM-TC1) and the spontaneously immortalized myoblasts (iBat-S1) were maintained in GM1 and subcultured at a 1:3–1:6 ratio every 2–3 days. Cells were cryobanked around every 10 passages up until passage 40 in freeze media prepared with 10% dimethyl sulfoxide (DMSO, Sigma) and 90% HyClone Characterized FBS. All cell lines tested negative for mycoplasma contamination.

### 2.5. Myotube Differentiation

For myotube differentiation, myoblasts were seeded with a cell number of 2 × 10^5^ per well in a 6-well cell culture plate (Corning) (or 1 × 10^5^ per well in a 24-well plate (IBIDI (Munich, Germany), Cat #: 82426) containing 3 mL growth media and incubated as described above until cell density reached >85% confluency. Then, growth media were replaced with different differentiation media (DM) to initiate differentiation (Table 3). Cell morphology changes were tracked daily following the initiation of differentiation.

### 2.6. Myoblast Proliferation Assay

Myoblast proliferation assay was performed as previously described [45]. Briefly, myoblasts were seeded with a cell number of 1–2 × 10^5^ per well in a 6-well cell culture plate containing 3 mL of different growth media: GM1, GM2, and GM3 (Table 2). Myoblasts were dissociated with TrypLE and counted every day until myoblasts were >90% confluent or began differentiation.

### 2.7. Chromosome Counting

Chromosome counting was performed as previously described [45]. Briefly, myoblasts were cultured in growth media in a T25 cell culture flask at 37 °C with 5% CO_2_, until cell density reached 60–80% confluency. Then cells were incubated in growth media containing 50 ng/mL colcemid (Invitrogen Life Technologies (Carlsbad, CA, USA), Cat#: 15212–012) at 37 °C with 5% CO_2_ for 2.5 h. Subsequently, myoblasts were washed with PBS, dissociated with TryplE, and collected through centrifugation at 300× *g* for 5 min. Cells were then resuspended in hypotonic solution (1 mL growth media and 9 mL 0.075 M KCl) (Gibco, Cat#:10575-090), incubated at 37 °C for 30 min, and centrifuged at 200× *g* for 5 min at RT. Eventually, the myoblasts were fixed through a gradual fixation method with freshly prepared ice-cold fixative (75% methanol, 25% glacial acetic acid). After which, cells were resuspended in 500 µL fixative and stored at 4 °C until use (usually less than 1 month). For the metaphase spreads preparation, the cell suspension was dropped onto a glass slide from a 10 cm distance and air-dried. Slides were mounted with VECTASHIELD^®^ Antifade Mounting Medium with DAPI (Vector Laboratories (Newark, CA, USA), Cat#: H-1200-10) and sealed with a coverslip. For chromosome counting, at least 50 metaphase spreads were analyzed using a fluorescence microscope. Image analysis and processing were performed using Fiji.

### 2.8. Immunofluorescence

Cells cultured in a 24-well IBIDI plate were washed with phosphate-buffered saline (PBS), fixed with 4% paraformaldehyde for 30 min at RT, washed in PBS, permeabilized for 30 min using 0.1% Triton-X, blocked for 60 min using SuperBlock™ Blocking Buffer (Thermo Scientific™ (Waltham, MA, USA), Cat #: 37580), and washed with PBST. For myoblast marker staining, primary PAX7 antibody (Santa Cruz (Dallas, TX, USA), Cat #: sc-81648) (1:50) and Primary Desmin Primary (Abcam (Waltham, MA, USA), Cat #: ab15200) (1:200) were diluted in Antibody Diluent Reagent Solution (Thermo Scientific™, Cat #: 003118), added to cells, and incubated overnight at 4 °C. Cells were then washed with PBST, and incubated with CF^®^488A Donkey Anti-Mouse IgG (H+L) (Biotium (Fremont, CA, USA), Cat #. 20014) and CF^®^647 Donkey Anti-Rabbit IgG (H+L) (Biotium, Cat #. 20047) (both diluted 1:500 in Antibody Diluent Reagent Solution) for 60 min at RT. After washing with PBST, the cell nuclei were stained with DAPI (Biolegend (San Diego, CA, USA), Cat # 422801) (used at 1:500 dilution in PBS) for 15 min at RT, then washed with PBS. For myotube marker MF20 staining, differentiated cells were fixed and stained as previously described, using primary MF20 antibodies (Invitrogen, Cat #: 14-6503-82) (1:200), CF^®^488A Donkey Anti-Mouse IgG (H+L) (Biotium, Cat #. 20014), and DAPI (Biolegend, Cat # 422801). For F-actin staining, the cells were stained with Phalloidin CF647 phalloidin conjugate (Biotium, Cat# 00041-T) (used 1:40 dilution in PBS) for 30 min at RT, then washed with PBS. Imaging was performed with a fluorescence microscope.

## 3. Results

### 3.1. Functional and Molecular Specializations Supporting Flight in Pteronotus parnellii

#### 3.1.1. Candidate Gene Expression Reveals Fiber-Type Heterogeneity

To investigate the structural and metabolic specializations of flight muscle, we first analyzed native pectoralis tissue using histological staining and RNA sequencing from *P. parnellii*. These data provide an in vivo reference for muscle phenotype and serve as a foundation for interpreting downstream cell-based assays. Histological and transcriptomic analyses revealed a mosaic fiber-type architecture in the pectoralis muscle (Figure 1). Periodic acid–Schiff (PAS) staining indicated heterogeneous glycogen distribution across fibers, consistent with a mixture of slow- and fast-twitch fiber types (Figure 1B). RNA-seq analysis confirmed co-expression of both fast-twitch and slow-twitch fiber markers (Figure 1C). Expression of fast-twitch genes such as *Myl1*, *Tnnt3*, *Myh1* (Type IIx), *and Myh4* (Type IIb) was strongly expressed. Slow-twitch markers, including *Myl3* and *Tnnt1* were also present at relatively high levels, revealing a subset of oxidative fibers. While *Myh7* (Type I) was not detected, paralog *Myh7b* was expressed at low levels. *Myh2* (Type IIa) was not detected. Additionally, calcium-handling genes showed mixed expression: *Casq1* and *Casq2* (fast and slow paralogs of calsequestrin) were expressed at similar levels, as were *Atp2a1* (*Serca1*) and *Atp2a2* (*Serca2*) (fast and slow paralogs of sarcoplasmic reticulum Ca^2+^-ATPase). These patterns support a predominance of fast-twitch glycolytic (Type IIb) and oxidative (Type IIx) fibers, alongside hybrid calcium-handling features involving both calcium release and reuptake mechanisms typical of fast- and slow-twitch muscle.

At baseline rest (0 min), flight muscle exhibits elevated expression of candidate genes involved in redox buffering and calcium handling, as defined by levels exceeding housekeeping gene expression (Appendix A). These include redox-related genes *Foxo3*, *Cat*, *Sod1*, and *Txnip*, as well as calcium-handling genes noted above (*Casq1*, *Casq2*, *Atp2a1*, and *Atp2a2).* While not differentially expressed across timepoints, their high expression suggests a constitutive role in supporting oxidative stability and calcium flux in muscle (Figure 1D). *Tmc5* was included as a calcium cycling candidate gene based on its putative functional role. Its low transcript abundance suggests it may play a more limited or context-specific role in muscle physiology (Appendix A). These findings highlight a predominant fast-twitch fiber composition with oxidative and calcium-handling adaptations, suggesting *Pteronotus* flight muscle is tuned for both high force production and sustained performance.

#### 3.1.2. DEG Analyses Reveal Flight Muscle Activation of Regeneration and Metabolism

To further characterize flight muscle specialization and its capacity to respond to induced metabolic activity, we examined gene expression changes following an acute glucose stimulation (5.4 g/kg body weight). We focused on differentially expressed genes (DEGs) between 30 and 60 min post-stimulation (Figure 1E), a window chosen to capture dynamic transcriptional responses to energy-induced stress. Using a cutoff of log_2_ fold change > 1.5 (Appendix A), we found that most DEGs were downregulated at 60 min relative to 30 min, reflecting transient early activation of muscle development and regeneration pathways. Gene set enrichment analysis identified significant over-representation of terms such as muscle cell differentiation (*p* = 5.2 × 10^−5^), muscle organ development (*p* = 7.8 × 10^−4^), and regeneration (*p* = 0.0024), consistent with an early activation of myogenic and structural programs. Conversely, genes upregulated at 60 min were enriched for terms related to metabolic processes, including the phosphatase complex (*p* = 6.7 × 10^−4^), manganese ion binding (*p* = 3.1 × 10^−5^), and muscle system process (*p* = 0.0012), suggesting a shift toward energy regulation and contractile function. Pathway-level enrichment analysis highlighted pathways relevant to muscle growth and glucose uptake including KEGG|PI3K-Akt signaling (*p* = 9.14 × 10^−3^), Panther | PI3 kinase pathway (*p* = 0.1188), and KEGG|focal adhesion (*p* = 1.83 × 10^−3^).

To visualize these time-dependent transitions, we generated a conceptual model overview (Figure 1F) integrating differentially expressed genes (30 and 60 min vs. baseline) and functionally relevant candidate genes with expression above the housekeeping baseline (Appendix A). Genes were grouped into functional modules reflecting redox buffering, nutrient and stress sensing (metabolic resilience), and regeneration. For example, markers of regeneration included DEGs such as *Hey2*, *Tmc5*, *Chad*, *Csrp3*, *Lingo4*, and *Myh4*, which peaked at 30 min and declined thereafter, while additional candidates with established roles in myogenic activation, cell cycle progression, and fiber-type specification (*Pax7*, *Mymx*, *Etf8*, *Tmeff1*, and *Maf*) were significantly upregulated compared to housekeeping gene levels (Appendix A). Genes associated with metabolic resilience, including DEGs *Etv5*, *Map4k3*, and *Slc4a7,* and genes involved in redox buffering (*Nrf1*, *Pak1*, and *Ucp2*) remained elevated. These dynamics suggest a coordinated transition from early muscle repair toward metabolic adaptation.

A direct comparison of 60 vs. 30 min post-stimulation (Figure 1G) confirmed this temporal shift, with many regeneration-related transcripts (e.g., *Mymk*, *Mymx*, *Tmc5*, and *Chrng*) decreasing while metabolic regulators (e.g., *Ppm1n*, *Ppp1r3g*) increased in expression. *Ppp1r3g*, a well-defined regulatory subunit of protein phosphatase 1, promotes glycogen synthase activation, and so its upregulation facilitates glucose storage in late-phase muscle activation. Notably, several highly upregulated genes at 60 min were uncharacterized loci (e.g., *LOC129064675*, *LOC129072723*, *LOC129068479*), potentially representing novel or bat-specific transcripts involved in late-phase muscle responses. Together, these results underscore a biphasic transcriptional response to glucose stimulation: an early activation of regeneration and structural programs followed by a later phase of metabolic reprogramming.

### 3.2. Isolation of PAX7^+^ Cells from Flight Muscle and Verification of Myogenic Cells

Given the transcriptional evidence for regenerative activation (Figure 1E), such as elevated expression of *Pax7* and *Hey2* following glucose stimulation (Appendix A), we hypothesized that flight muscle harbors a large resident population of muscle stem cells (satellite cells) that could be isolated and expanded in vitro. To test the cellular basis of gene expression patterns identified in *P. parnellii*, we adopted a pre-plating strategy commonly used for myoblast enrichment in mammalian systems [81,82,83,84]. Using pre-plating, we isolated stable, self-renewing satellite cells from the pectoralis major muscle (Figure 2A). Although we initially collected *P. parnellii* muscle tissue, it was not preserved in a way that allowed for viable cell isolation, as methods for culturing bat myoblasts were still under development. By the time our protocol was optimized, institutional travel restrictions prevented a return to Trinidad, currently a level III travel advisory, where *P. parnellii* is found. We therefore selected *P. mesoamericanus*, a closely related species with similar ecology and flight performance, making it a suitable comparative model species and biologically relevant alternative for in vitro studies.

Primary cells began attaching to the cell culture dish 3–4 days after seeding (Figure 2B). By day 6, cultures reached 80% confluence with a mix of spindle- and triangle-shaped cells (Figure 2B). After 3 rounds of pre-plating, the adherent cell population became uniform in morphology (Figure 2C), typical of activated satellite cells and early myoblasts, suggesting successful enrichment of myogenic progenitors. Immunofluorescence staining confirmed co-expression of the muscle stem cell marker PAX7 and the intermediate filament protein DESMIN, validating the presence of activated muscle stem cells (Figure 2D). In addition, when switched to growing in standard differentiation media (DM1), these cells readily fused into multinucleated myotubes within 48 h (Figure 3A). Myotube identity was confirmed by immunostaining for myosin heavy chain (MyHC), a myotube marker (Figure 3B,C). Interestingly, many differentiated myotubes exhibited frequent spontaneous contraction as early as two days post-differentiation (Appendix A), indicating functional integrity and neuromuscular activity.

### 3.3. Establishment of Immortalized Bat Myoblast (iBatM) Cell Lines

Following successful isolation and validation of primary myogenic progenitors, we next aimed to establish long-term expandable muscle cell lines. To establish immortalized bat cell lines, primary *P. mesoamericanus* myoblasts were co-transduced with lentiviral constructs expressing *hTERT* and *CDK4* genes, each containing a blasticidin selection marker. The resulting engineered line, iBatM-Pmeso-TC, showed stable proliferation over multiple passages (Appendix A).

In parallel, a separate population of non-transduced primary cells entered a transient proliferative crisis before recovering and resuming growth, giving rise to a spontaneously immortalized line, iBatM-Pmeso-S. Both lines have been maintained for over 50 passages across 5 months in culture, with no observable changes in cellular morphology and differentiation capability (Appendix A). These results confirm successful immortalization by both engineered and spontaneous mechanisms.

To assess whether these lines retained myogenic identity, we performed immunofluorescence staining for the canonical markers PAX7 and DESMIN. The hTERT/CDK4-immortalized iBatM-Pmeso-TC line co-expressed both markers at passage 37 (Figure 2F), consistent with an activated satellite cell state, similar to primary cell cultures (Figure 2D). In contrast, the spontaneous line iBatM-Pmeso-S retained DESMIN expression with a low or absent PAX7 population (Figure 2E), indicating a more committed myoblast phenotype with reduced stemness. These results demonstrate that the two immortalized bat cell lines can be derived through two mechanisms and that they capture distinct stages of the myogenic lineage, providing flexible and renewable in vitro models for muscle biology and regeneration.

### 3.4. Immortalized Bat Myoblasts Are Genetically Stable

To validate the genomic stability of the immortalized lines, we performed karyotype analysis on both primary and immortalized long-term passaged cells. Primary *P. mesoamericanus* myoblasts at passage 9 (P9) exhibited a diploid chromosome number of 38 (2N = 38), consistent with our previous karyotyping of bat fibroblasts [45] and published karyotypes for *Pteronotus* species [85] (Figure 4A). Both immortalized cell lines, the engineered iBatM-Pmeso-TC and the spontaneous iBatM-Pmeso-S, retained the same diploid count at passage 37 (P37) (Figure 4B,C), indicating that extended passaging did not compromise chromosomal integrity. These results confirm that the immortalized bat myoblasts remain genetically stable over time, supporting their use as a reliable in vitro platform.

### 3.5. Immortalized Bat Myoblasts Retain the Proliferation Capacity of Bat Primary Myoblasts

To evaluate the long-term proliferative stability of the immortalized lines, we measured doubling time at early (P8), mid (P20), and late (P40) passages. Both iBatM-Pmeso-TC and iBatM-Pmeso-S proliferated well in standard F10-based growth medium (GM1) across all timepoints (Figure 5 and Appendix A). At P8, doubling times were 23.42 h for iBatM-Pmeso-TC and 26.33 h for iBatM-Pmeso-S (Figure 5A,B). At P20, both cell lines showed slower proliferation: 31.07 h for iBatM-Pmeso-TC and 37.22 h for iBatM-Pmeso-S, followed by recovery at P40 (25.46 h and 28.96 h, respectively; Figure 5A,B). These trends indicate that both cell lines maintain stable growth profiles after the initial adaptation to culture, with rates comparable to primary bat myoblasts.

We next tested how basal media composition affects cell proliferation and differentiation potential. Previous studies have shown that different growth media can influence myoblast behavior, including stemness maintenance and initiation of differentiation [79,86,87]. To systematically assess this in bat myoblasts, both cell lines were cultured in three growth media formulations: F10 (GM1), DMEM/F10 (1:1; GM2), or DMEM (GM3), each supplemented with identical growth factors and serum (Table 2). Morphological assessment revealed no notable differences across conditions (Appendix A), but cell counts showed that both cell lines proliferated more rapidly in GM2 and GM3 compared to GM1 (Appendix A).

In addition, cells in GM2 and GM3 began differentiating into myotubes once cultures reached >95% confluence, with widespread myotube formation observed by day 3 and detachment by day 8 (Appendix A). In contrast, cells maintained in GM1 remained largely undifferentiated even on day 8 under high-density conditions, showing minimal myotube formation. These results suggest that bat myoblast lines retain long-term proliferative capacity and respond to media composition in a predictable manner, growing faster and initiating differentiation in DMEM-based media, while maintaining a more undifferentiated, stem-like state in F10-based conditions.

### 3.6. Immortalized Bat Myoblasts Retain the Differentiation Capacity of Bat Primary Myoblasts

To evaluate whether the immortalized lines retained their capacity to differentiate into functional muscle fibers, iBatM-Pmeso-TC and iBatM-Pmeso-S were cultured in standard differentiation media (DM1). Both cell lines successfully formed multinucleated myotubes within 2 days (Appendix A), and spontaneous myotube contractions were observed by day 3 (Appendix A). Immunofluorescence staining confirmed expression of MyHC (MF20), validating their myogenic identity (Figure 3D,E). The F-actin staining revealed differences in cytoskeletal organization: iBatM-Pmeso-TC myotubes exhibited partially aligned F-actin bundles resembling sarcomeric organization, whereas myotubes differentiated from iBatM-Pmeso-S cells and iBatM-Pmeso-TC displayed punctate, irregular F-actin with limited structural alignment (Figure 3D,E), suggesting that iBatM-Pmeso-S cells have reduced cytoskeletal maturation [88]. Despite this, the iBatM-Pmeso-S line formed significantly more myotubes under the same conditions, indicating greater fusion efficiency but potentially limited contractile organization. These differences highlight variation in functional maturation between the two lines, possibly reflecting immortalization strategy or progenitor content.

To further examine how culture conditions influence differentiation, we systematically tested four differentiation media formulations (DM1-DM4; Table 3) that varied in basal media (DMEM vs. F10). Both cell lines formed mature multinucleated myotubes in DMEM-based media (DM1 and DM3), but not in F10-based media (DM2 and DM4), even with extended culture (Appendix A). This finding is consistent with our previous observation that F10 supports a more stem-like, undifferentiated state, while DMEM promotes differentiation.

Since myoblasts fuse into myotubes in GM3, we further evaluated the effects of serum (2% horse serum versus 20% HyClone Characterized FBS) on bat myotube differentiation. Cells cultured in GM3, DM1, and DM3 formed functional myotubes, while no myotube formation was seen in GM1, DM2, and DM4 (Appendix A). This suggests that serum reduction is not required for differentiation in these bat myoblast lines and that basal media composition is the primary driver of differentiation competence. Together, these findings demonstrate that both immortalized myoblast lines retain functional differentiation capacity and contractile potential, and that differentiation efficiency is strongly affected by basal media composition. Building on these results, we next examined whether differentiated myotubes exhibited functional behaviors reflective of mature muscle, such as spontaneous contraction.

### 3.7. Immortalized Bat Myoblasts Enable Functional Profiling of Flight Muscle Under Metabolic Overload

The establishment of spontaneously contracting myotubes from both primary (Appendix A) and immortalized cells (Appendix A) offers a unique in vitro system to investigate how bat skeletal muscle supports the extreme physiological demands of powered flight. While detailed mechanistic assays are ongoing, the robust and early-onset contractile activity observed during differentiation suggests enhanced functional maturation, potentially involving neuromuscular junction (NMJ) priming or sustained excitation–contraction coupling. Such spontaneous contraction is rarely observed under standard culture conditions (Table 4), underscoring the distinctive physiology of bat myotubes. To explore whether these in vitro findings reflect physiological traits in vivo, we next examined skeletal muscle function and energy storage in *Pteronotus*, integrating glucose dynamics, biochemical measures of muscle energetics, and transcriptomic data.

To contextualize this cellular behavior within a physiological context, we examined the whole-animal metabolic response to glucose overload in *P. mesoamericanus* and *P. parnellii* using the standardized glucose tolerance tests (GTT), a non-lethal assay of systemic metabolic function [75]. Following the glucose challenge, *P. mesoamericanus* maintained tightly regulated blood glucose levels, indicating robust systemic glucose regulation (Figure 6). In contrast, the GTT for *P. parnellii* showed broader inter-individual variation, with several individuals exhibiting hyperglycemia beyond 300 mg/dL at 60 min post-feeding (Figure 6B). To highlight this variation, individuals were grouped post hoc into two phenotypes based on glucose clearance: those with efficient regulation (gray) and those with hyperglycemia (red). While these groupings were used for visualization, the overall species-level response was modeled using a linear mixed-effects approach incorporating all individuals. Although *P. parnellii* showed greater within-species variation than other taxa, we present a single model for *P. parnellii* to capture the overall species-level response; the sample size per clearance phenotype was not sufficient to support robust subgroup modeling. The model confirmed significant species-level differences in glucose clearance (*p* < 0.001; Figure 6B).

While GTTs provided a non-lethal readout of systemic glucose regulation in both *P. mesoamericanus* and *P. parnellii*, only the latter permitted paired tissue sampling to examine underlying muscle biochemistry. During flight, skeletal muscle is the dominant consumer of glucose and ATP, making it a key tissue for buffering metabolic fluctuations. To connect systemic glucose handling with tissue-level muscle physiology, we analyzed previously collected *P. parnellii* samples to assess energy storage strategies in flight muscle. Biochemical profiling revealed that energy reserves were primarily stored as triglycerides, with relatively lower levels of glycogen (Figure 6A), consistent with a lipid-based endurance metabolism that may support flight endurance and buffer against glucose fluctuations. This may help explain the sustained hyperglycemia observed in some individuals following a glucose challenge (Figure 6B), suggesting inter-individual variation in muscle fuel handling or metabolic flexibility.

To identify the molecular programs that may underlie variation in glucose handling and energy storage, we returned to transcriptomic analysis of *P. parnellii* pectoralis muscle collected 30 to 60 min after glucose stimulation. Differential expression analysis (|log_2_FC| > 1.5, FDR < 0.01) revealed genes associated with fuel utilization and stress response (Appendix A). Functional clustering of enriched pathways, based on Jaccard similarity among g:Profiler-enriched terms, grouped DEGs into four major functional modules: glucose utilization, lipid handling, metabolic signaling, and flight endurance (Figure 6C). Key lipid-handling genes such as *Fabp2* and *Etnppl* suggest active beta oxidation and mitochondrial membrane integrity during triglyceride mobilization. Endurance-related genes *Chrnd* and *Colq*, as well as contractile genes *Myh4* and *Csrp3*, support contraction-coupled glucose uptake.

A Venn diagram illustrating gene assignments across modules showed substantial overlap (Figure 6D), underscoring the coordinated nature of muscle responses to nutrient load. Notably, signaling genes *Map4k3* (nutrient sensing) and *Osmr* (cytokine-mediated) were uniquely assigned, pointing to upstream metabolic and regenerative processes. To further contextualize molecular findings, we developed a schematic model of core cellular pathways supporting flight muscle performance (Figure 6E). Upstream metabolic and cytokine signaling pathways act through receptors IRS1/2 and OSMR, both of which were transcriptionally upregulated following glucose stimulation. These signals converge on nutrient-sensitive regulators such as MAP4K3 and downstream effectors mTOR and JNK, which mediate stress and metabolic responses. Within this postprandial state, pyruvate-derived acetyl-CoA is directed into the TCA cycle or diverted toward triglyceride (TG) synthesis, aligning with our biochemical data showing TG-enriched flight muscle.

Lipid-driven ATP production may support prolonged muscle contraction and stable neuromuscular junction (NMJ) signaling, which are likely prerequisites for endurance flight. This is evidenced by the upregulation of postsynaptic genes (*Chrnd*, *Colq*) and key contractile regulators (*Myh4*, *Csrp3*). In parallel, *Osmr* contributes to regenerative signaling through activation of muscle stem cell pathways, consistent with our earlier evidence of PAX7^+^ progenitor activation. By synthesizing cellular phenotypes, physiological assays, and dynamic transcriptional responses, this multi-scale framework highlights how bat flight muscle coordinates metabolic flexibility, contractile function, and stem cell activity to sustain muscle performance.

## 4. Discussion

Skeletal muscle is a metabolically dynamic tissue essential for locomotion, energy homeostasis, and repair. In flying mammals, these demands are intensified by the energetic and mechanical load of powered flight. To investigate how bat muscle accommodates such stress, we first performed transcriptomic profiling of *Pteronotus* pectoralis muscle before and after glucose stimulation—a physiological proxy for flight energy overload (Figure 1). This in vivo analysis revealed enriched expression of genes involved in calcium handling (e.g., *Casq1*, *Atp2a1*), redox buffering (*Foxo3*, *Cat*, *Sod1*), membrane transport (*Slc16a1*), and nutrient sensing (*Igf1*, *Nr1d1*), as well as neuromuscular components (*Chrnd*, *Colq*) and contractile regulators (*Myh4*, *Csrp3*). These findings define molecular programs that support energy production, oxidative resilience, and sustained contraction under metabolic load, laying the foundation for understanding muscle performance in the context of flight. To enable direct experimental access to these mechanisms, we next established the first immortalized bat myoblast cell lines. These lines serve as a renewable in vitro platform to study muscle development, regeneration, and metabolism in a species naturally adapted for powered flight.

Myoblasts, the progenitor cells of skeletal muscle, are essential for muscle development, injury repair, and regeneration [2,3]. In the present study, we report the first successful establishment of primary myoblast cells from bats. These bat cells retain hallmarks of myogenic identity, including PAX7 and DESMIN protein markers, and readily differentiate into multinucleated myotubes. Remarkably, the differentiated myotubes exhibited spontaneous contractions in vitro, suggesting intrinsic properties of excitability and contractile readiness that may reflect specialized flight muscle physiology (Appendix A). While these observations point to functional maturation of bat myotubes, future work will incorporate quantitative measures such as contraction frequency, calcium imaging, and electrophysiological recordings to characterize excitation-contraction coupling in these cells.

Isolating primary myoblasts from pectoralis muscle is technically challenging due to the mix of myoblasts, fibroblasts, and other differentiated cell types. Fibroblasts and myoblasts are both proliferative and share morphology, which complicates enrichment, particularly in non-model species lacking validated markers and reagents. To address this, we used a pre-plating method based on differential adhesion to enrich for myoblasts, following protocols established in other systems [81,82,83,84].

Building on this enrichment strategy, we successfully established two immortalized bat myoblast cell lines: iBatM-Pmeso-S through spontaneous immortalization and iBatM-Pmeso-TC via *hTERT/CDK4* gene overexpression. These strategies are informed by prior models (Table 1) such as L6 cells [5], C2C12 [6], hTERT-immortalized myoblasts from human [7,8,11,57] and canine sources [10], and spontaneous immortalization of fish myoblasts [12,15,93]. As with other hTERT/CDK4-immortalized models, genomic integration may occur randomly, posing potential risks of transformation or altered gene regulation. Both bat cell lines retain myogenic potential and stable karyotypes, similar to primary cells, with doubling times of ~25–29 h at passage 40. Notably, iBatM-Pmeso-TC preserved a higher proportion of PAX7^+^ progenitor cells than iBatM-Pmeso-S and even primary myoblasts, suggesting that *hTERT/CDK4* gene overexpression may enhance stemness by bypassing senescence pathways.

Having two immortalized cell lines that differ in progenitor maintenance provides a valuable framework for studying myogenic states and their regulation under various metabolic or physiological stresses. A key marker of myotube maturation is F-actin organization, which reflects cytoskeletal alignment and sarcomere assembly in differentiated myotubes. Under identical staining conditions, both primary cells and iBatM-Pmeso-TC myotubes displayed the expected striated F-actin patterns, consistent with mature contractile architecture. In contrast, iBatM-Pmeso-S myotubes showed punctate and irregular F-actin staining, suggesting disrupted cytoskeletal organization. These structural observations are consistent with reduced myogenic identity or impaired differentiation capacity in the spontaneously immortalized line, possibly linked to the lower proportion of PAX7^+^ cells in iBatM-Pmeso-S. The preservation of a more uniform F-actin organization in iBatM-Pmeso-TC supports the use of targeted hTERT/CDK4 immortalization to retain key features of muscle cell maturation. Future work will map integration sites to help assess the safety, reproducibility, and stability of the model, particularly for endogenous gene regulation and cellular behavior over time.

Despite reduced cytoskeletal organization, iBatM-Pmeso-S myotubes retained the ability to undergo spontaneous contractions, indicating that basic excitation–contraction coupling can occur in the absence of fully aligned sarcomeres. This uncoupling between structural maturation and functional output may model adaptive or stress-resilient muscle states and could be relevant for studying minimal requirements for contractility in contexts such as muscle regeneration, aging, or metabolic dysfunction. These differences in structural and functional maturation between the two immortalized lines prompted us to examine whether external cues such as basal media composition could further influence differentiation outcomes and cytoskeletal organization.

We optimized cell culture media to balance proliferation and differentiation based on insights from other systems [15,94,95]. Growth media supplemented with fibroblast growth factor (FGF), epidermal growth factor (EGF), insulin, and dexamethasone media might promote robust proliferation and stemness preservation. Among the basal media tested, F10-based media were most effective at maintaining undifferentiated progenitor states, while DMEM promoted myotube differentiation. The contrast is likely driven by DMEM’s higher concentrations of glucose (4.5 g/L vs. 1.1 g/L in F-10), as glucose restriction below 0.9 g/L has been shown to block myotube formation [73,96]. DMEM also contains calcium and essential amino acids (leucine, valine, and lysine), factors known to promote mTOR signaling and myoblast fusion [97,98]. In contrast to DMEM media, F10 instead contains a broader range of micronutrients and amino acids that support stemness, including copper, zinc, hypoxanthine, lipoic acid, alanine, asparagine, aspartic acid, glutamic acid, proline, biotin, and vitamin B12 [99,100,101]. These findings emphasize the importance of species-specific formulations and suggest that F10 media may be well suited for regenerative studies, while DMEM is more appropriate for differentiation protocols.

Importantly, we found that base media composition had a stronger effect on differentiation than serum concentration. While low serum conditions are typically used to induce myotube differentiation in vitro [15,57,87,102], our results show that myoblasts cultured in F10-based media failed to differentiate even under low serum conditions. In contrast, cells in DMEM or DMEM/F10 rapidly formed functional myotubes regardless of serum levels. This suggests that nutritional cues (e.g., glucose and amino acids) may be a primary driver of myotube fusion in bat myoblasts, with implications for interpreting metabolic signaling during bat muscle development. In future work, we plan to evaluate additional molecular markers of terminal differentiation to define more precisely the extent of maturation achieved by bat myotubes under different culture conditions. This will include analysis of myogenin (Myog), which marks entry into the differentiation program, and Myf6 (MRF4), a late-stage myogenic regulator associated with terminal differentiation and maintenance of mature muscle fibers. We also intend to assess adult MyHC isoforms and sarcomeric structural proteins to determine whether functional contractility in bat myotubes is accompanied by molecular hallmarks of terminal muscle identity. To further explore the molecular programs that support contractile function in bat muscle, we next analyzed RNA-seq data from native pectoralis tissue to identify gene expression patterns associated with excitation–contraction coupling and energy metabolism during muscle activation.

The transcriptomic profiling observed in native *Pteronotus* pectoralis tissue closely aligns with our in vitro findings, supporting the physiological relevance of the cell-based model. Upregulated genes involved in calcium handling (*Casq1*, *Casq2*, *Atp2a1*, *Atp2a2*), redox buffering (*Foxo3*, *Cat*, *Sod1*), membrane transport (*Slc16a1*), nutrient sensing (*Igf1*, *Nr1d1*), and contractile function (*Myh4*, *Csrp3*) mirror the functional phenotype of spontaneous myotube contraction. Additionally, several of these genes exhibited high expression at baseline (0 min), suggesting a constitutive role in supporting oxidative stability, calcium flux, and contractile readiness—features essential for the demands of powered flight. A systems-level model (Figure 6E) integrates these gene expression signals into a functional protein interaction framework, emphasizing how intracellular components of nutrient and cytokine signaling pathways contribute to muscle function. In the model, transcriptional upregulation of signaling intermediates such as OSMR and MAP4K3 reflects the activation of downstream pathways linked to Oncostatin M (OSM) and nutrient sensing. While expression of canonical insulin signaling intermediates such as *Irs1* and *Irs2* was limited in our dataset, insulin remains a relevant upstream regulator. Instead, we observed strong early upregulation of *Osmr*, supporting a role for OSM signaling in coordinating regenerative responses [103]. This is consistent with findings from Sampath et al. [104], who identified OSM as a potent niche-derived cytokine that induces reversible quiescence in muscle stem cells while preserving regenerative capacity. In this context, *Osmr* expression in bat muscle may reflect a mechanism for balancing regenerative activation with maintenance of stemness under metabolic load. Both nutrient and cytokine pathways converge on downstream effectors such as mTOR and JNK, linking upstream signaling inputs to metabolic reprogramming and contractile readiness.

In addition to identifying key functional pathways, our results reveal a biphasic temporal response to glucose stimulation. Early transcriptional activation includes genes associated with regeneration and structural remodeling (e.g., *Mymx*, *Tmc5*, *Hey2*, *Chad*), while later expression shifts toward metabolic regulators (*Ppp1r3g*, *Etnppl*, *Fabp2*). This sequential response is consistent with time-resolved transcriptomic data from human skeletal muscle. For example, Pillon et al. [105] showed that acute exercise triggers early signaling events followed by metabolic programs in myosin isoform expression and muscle structural remodeling. Similarly, Robinson et al. identified distinct transcriptional waves following exercise, with early activation of immune-related genes and delayed enrichment of metabolism pathways [106]. A comparable biphasic transcriptional profile was observed by Kokaji et al. who used time-resolved transcriptomic analysis to show that oral glucose stimulation in mice induces early signaling gene expression followed by later metabolic reprogramming in skeletal muscle [107]. Together, these studies support a conserved temporal framework in which muscle integrates early signaling or stress responses with delayed metabolic adaptation. Our findings extend this model to a flight-adapted mammal, demonstrating that the *Pteronotus* muscle exhibits sequential transcriptional phases following glucose stimulation. This approach serves as a physiological proxy for flight or exercise, as glucose influx represents a key metabolic trigger for muscle activation, energy storage, and utilization under high-performance demands.

These transcriptomic and functional data validate the physiological relevance of our immortalized cell lines and provide the foundation for future mechanistic studies for flight muscle biology. These data lay the foundation for downstream experimental studies—including metabolic flux (assays of glucose and fatty acid oxidation), mitochondrial dynamics, and insulin-independent glucose uptake—to directly test hypotheses about metabolic specialization in flying mammals. More broadly, the availability of these renewable lines now enables a unique opportunity to investigate lineage-specific muscle adaptations in a clade of mammals that remain largely unexplored in vertebrate muscle biology.

Finally, the establishment of these bat myoblast lines provides a new experimental system alongside standard rodent lines such as C2C12 and L6 (Table 4). While rodent models have been invaluable for studying muscle differentiation and insulin signaling, they originate from short-lived terrestrial, glycolytic mammals with limited oxidative capacity. Moreover, both C2C12 and L6 cells are known to stall at intermediate stages of myogenic differentiation and rarely progress to the terminal maturation steps observed in vivo. In contrast, *P. mesoamericanus* is a flight-adapted species with high muscular endurance, physiological resilience, and longevity. Our bat myotubes express sarcomeric myosin heavy chain (MF20) and exhibit spontaneous contractions, suggesting they may achieve a more advanced state of functional differentiation. The development of these stable bat cell lines enables comparative studies to investigate conserved and lineage-specific muscle adaptations, with potential relevance to aging, metabolic disease, and therapeutic muscle repair.

## 5. Conclusions

In this study, we isolated primary myoblasts from wild-caught bat pectoralis muscle and successfully established two genetically stable myoblast cell lines. This is the first foundational in vitro platform to study muscle regeneration and function in a flying mammal. Both immortalized bat cell lines retain proliferative capacity and readily differentiate into functional myotubes, exhibiting spontaneous contraction. Comparative optimization of culture conditions revealed that DMEM supports myoblast proliferation and myotube differentiation, while F10 media more effectively preserves the progenitor state, offering flexibility for applications focused on muscle repair vs. maintenance. Finally, by integrating in vitro cell behavior, in vivo physiology, and transcriptomic profiling, we uncovered a biphasic muscle response to metabolic overload in bats, characterized by early regenerative activation, sustained contractile readiness, and a reliance on triglyceride-based energy stores. By capturing these coordinated responses across scales, the iBatM cell lines offer a powerful system to investigate both conserved and lineage-specific strategies that support muscle performance under extreme physiological stress and evolutionary adaptations for powered flight.

## Figures and Tables

**Figure 1 cells-14-01190-f001:**
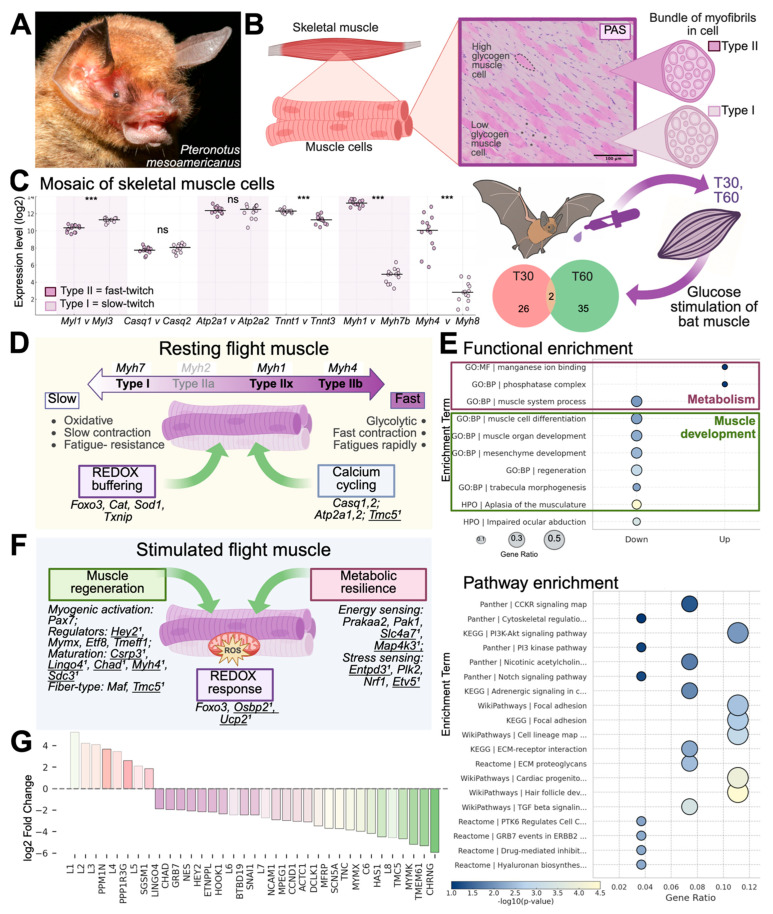
*Pteronotus* as a model for flight muscle specialization and stress resilience. (**A**) Photo of *P. mesoamericanus*, a Neotropical mustached bat specialized for high-speed, high-endurance flight. (**B**) Schematic of skeletal muscle highlighting a mosaic of fast-twitch (Type II) and slow-twitch (Type I) muscle fibers. Periodic Acid-Schiff (PAS) staining reveals high and low glycogen-content across muscle fibers. Skeletal muscle tissue (pectoralis) was sampled at three timepoints: control animals (*n* = 4) represent the baseline state (T0), while glucose-fed animals were sampled at 30 (T30; *n* = 3) and 60 (T60; *n* = 5) minutes post-glucose intake. RNA-seq was performed on flight muscle to identify differentially expressed genes (DEGs) in response to glucose. The Venn diagram shows overlap of DEGs between T30 vs. T0 and T60 vs. T0. (**C**) Expression of muscle fiber-type genes across 13 individuals, comparing fast-twitch (dark purple) and slow-twitch (light purple) paralogs. Candidate genes include myosin heavy chains (*Myh1*, *Myh4*, *Myh7*, *Myh8*), myosin light chains (*Myl1*, *Myl3*), calsequestrins (*Casq1*, *Casq2*), calcium ATPases (*Atp2a1*, *Atp2a2*), and troponin T (*Tnnt3*, *Tnnt1*). Horizontal line indicates median; statistical comparisons were performed using paired *t*-tests. *** *p* < 0.001. (**D**) Summary diagram of resting flight muscle gene expression at baseline. The pectoralis has mosaic fiber types, ranging from slow-twitch oxidative (Type I, *Myh7*) to fast-twitch glycolytic (Type IIb, *Myh4*), and intermediate fast-twitch oxidative-glycolytic fibers (Type IIx, *Myh1*). At rest, flight muscle is enriched for genes involved in redox buffering (*Foxo3*, *Cat*, *Sod1*, *Txnip*) and calcium handling (*Casq1/2*, *Atp2a1/2*, *Tmc5*^†^). ^†^ indicates DEGs with *p* < 0.003 and log_2_ fold change > 1.5. (**E**) Gene set enrichment analysis (GSEA) for T30 vs. T60. Dot plots show top enriched terms from KEGG, Reactome, Panther, and WikiPathways. Circle size reflects gene ratio; color indicates −log10 (*p*-value). Directionality refers to changes relative to T30: “Up” = higher expression in T60, “Down” = lower. The top functional terms were associated with muscle metabolism and development. (**F**) Summary diagram of glucose-stimulated gene expression, grouped by functional modules. Regeneration signaling includes myogenic activation (*Pax7*), regulators (*Mymx*, *Hey2*^†^, *Etf8*, *Tmeff1*), maturation (*Lingo4*^†^, *Chad*^†^, *Csrp3*^†^, *Myh4*^†^), and fiber-type specification (*Maf*, *Tmc5*^†^). Metabolic resilience includes energy sensing (*Prkaa2*, *Pak1*, *Slc4a7*^†^, *Map4k3*^†^) and stress signaling (*Etv5*^†^, *Plk2*, *Nrf1*). Redox response includes *Foxo3*, *Osbp2*^†^, and *Ucp2*^†^. (**G**) Log_2_ fold change in gene expression for T60 relative to T30. All data marked with † are significant from RNA-seq differential expression analysis. *n* denotes number of biological replicates. ns = not significant.

**Figure 2 cells-14-01190-f002:**
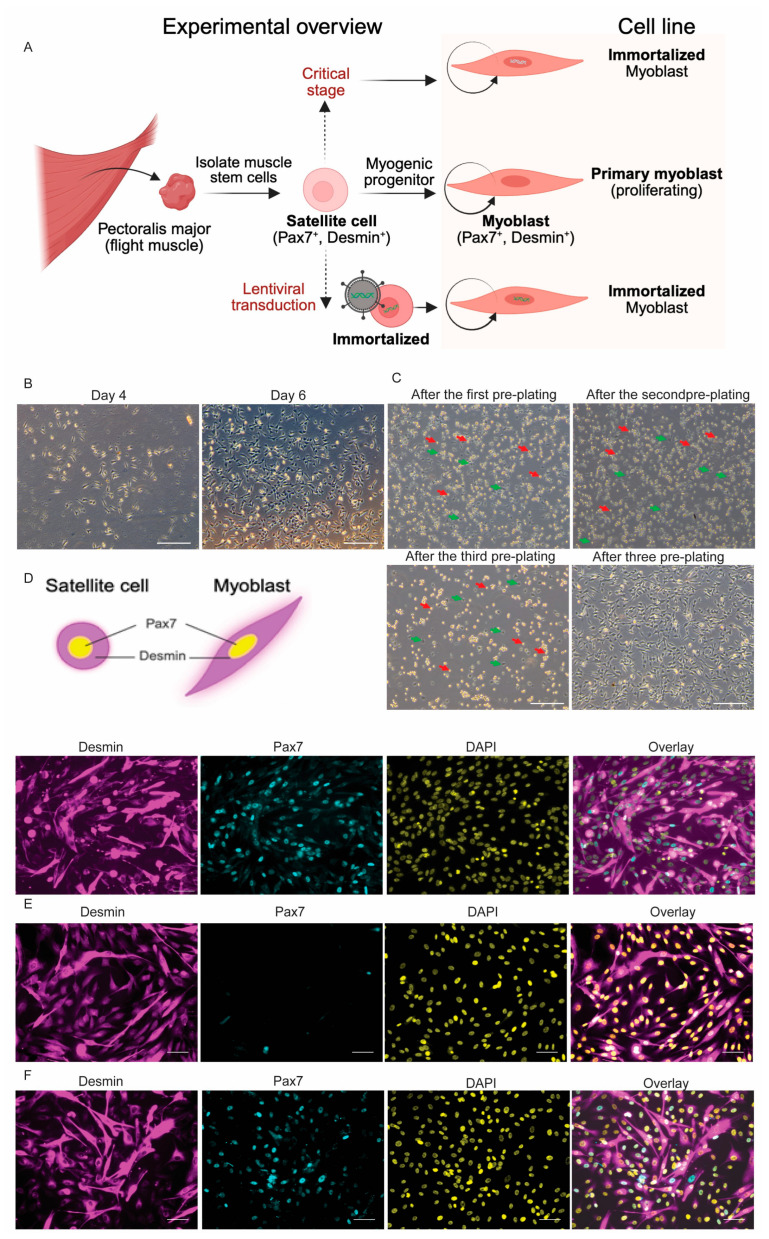
Bat primary myoblast isolation and myoblast verification. (**A**) Summary of the development and characterization of myoblast cell lines. (**B**) Primary muscle cells at 4-day and 6-day post-derivation. (**C**) Representative phase-contrast images showing the muscle cells at different stages of pre-plating. Green arrow: attached cells; red arrow: cells in suspension. (**D**–**F**) Immunofluorescent staining of DESMIN (magenta) and PAX7 (cyan) with DAPI counterstain (yellow) of *P. mesoamericanus* primary myoblasts, P9 (**D**), iBatM-Pmeso-S, P37 (**E**), and iBatM-Pmeso-TC, P37 (**F**). Scale bar: 50 μm.

**Figure 3 cells-14-01190-f003:**
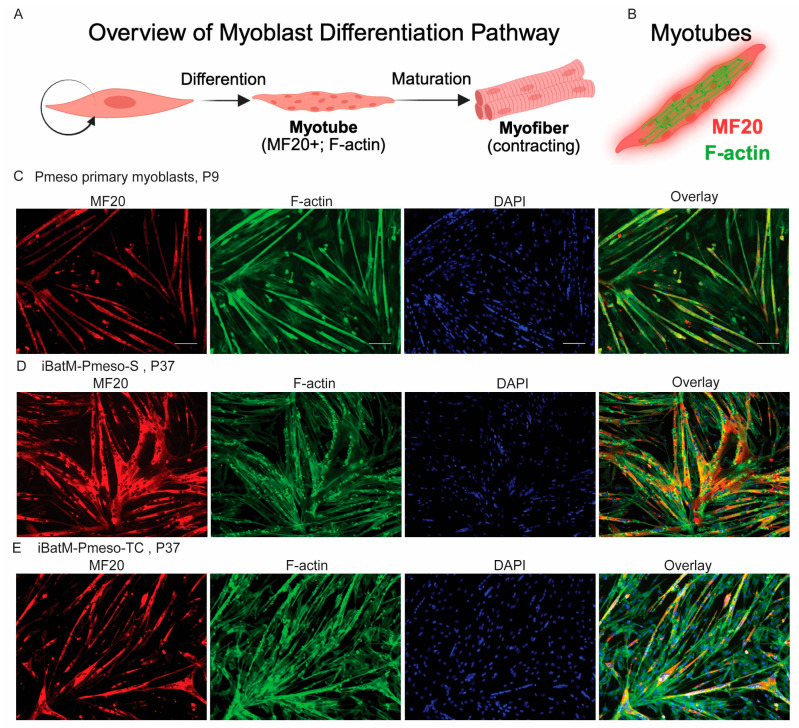
Bat Myoblast Lines Retain Differentiation Capacity. (**A**) Schematic overview of myoblast differentiation into multinucleated myotubes and subsequent maturation into myofibers. (**B**) Diagram illustrating expected colocalization of sarcomeric myosin heavy chain (MF20, red) and F-actin (green) in differentiated myotubes. (**C**–**E**) Immunofluorescent staining of myosin heavy chain (MF20; red), F-actin (green), and nuclei (DAPI; blue), with merged overlays to the right. (**C**) *P. mesoamericanus* myoblasts at passage 9 (P9) exhibit robust differentiation with organized, striated F-actin consistent with sarcomeric structure. (**D**) iBatM-Pmeso-S (P37) displays irregular, punctate F-actin patterns with less consistent alignment, suggesting impaired cytoskeletal maturation despite MF20 expression. (**E**) iBatM-Pmeso-TC (P37) shows improved F-actin organization and greater alignment compared to iBatM-Pmeso-S, more closely resembling the sarcomeric pattern observed in primary cells. Scale bar: 100 μm.

**Figure 4 cells-14-01190-f004:**
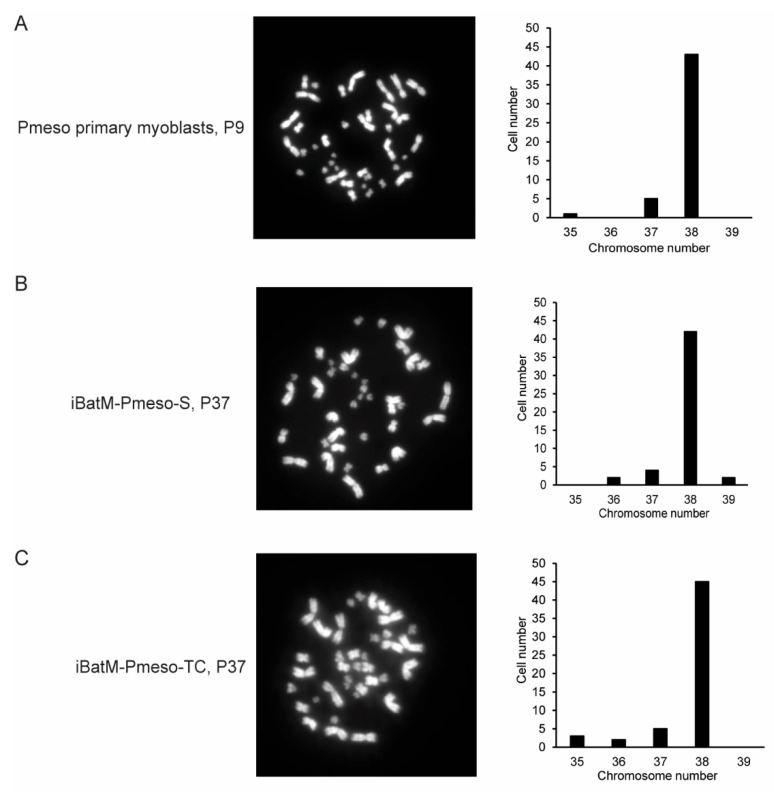
Bat Myoblast Lines Retain Genetic Stability After Extended Passaging. (**A**). Representative chromosome spread and count distribution for *P. mesoamericanus* primary myoblasts (Pmeso) at passage 9. A total of 49 metaphase spreads were analyzed, with a diploid karyotype of 2n = 38. (**B**). Representative chromosome spread and counts for iBatM-Pmeso-S at late passage (P37). Of 50 metaphase spreads, most maintained the expected diploid number (2n = 38), indicating genomic stability despite extended culture. (**C**). Representative image of chromosome spread and chromosome counts for iBatM-Pmeso-TC at P37. Among 55 metaphase spreads, the majority maintained 2n = 38, consistent with long-term karyotypic stability. Images were taken at 40× magnification.

**Figure 5 cells-14-01190-f005:**
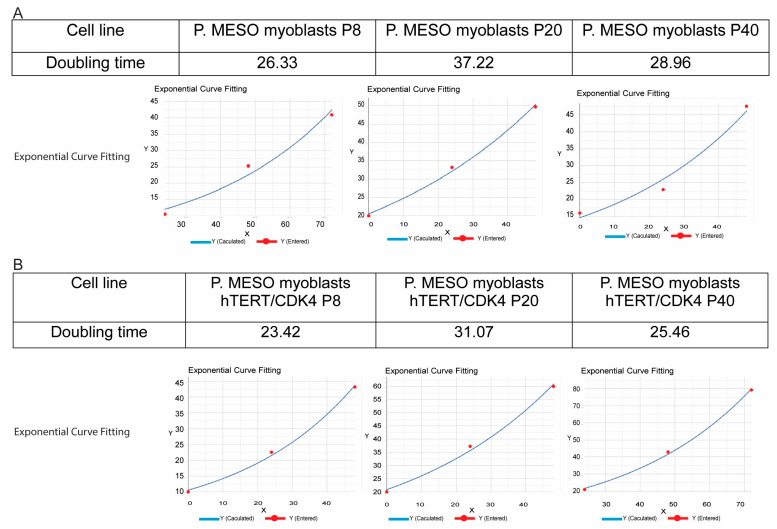
Spontaneous and hTERT/CDK4 Immortalization Support Long-Term Proliferation. (**A**). Doubling time and exponential curve fitting for spontaneously immortalized myoblasts iBatM-Pmeso-S at passages 8 (P8), P20, and P40, demonstrating the bypass of senescence. (**B**). Doubling time and exponential growth curve fitting for hTERT/CDK4-immortalized myoblasts iBatM-Pmeso-TC at P8, P20, and P40, showing stable proliferation. Doubling Time Computing software: https://www.doubling-time.com/compute_more.php (accessed on 5 May 2025).

**Figure 6 cells-14-01190-f006:**
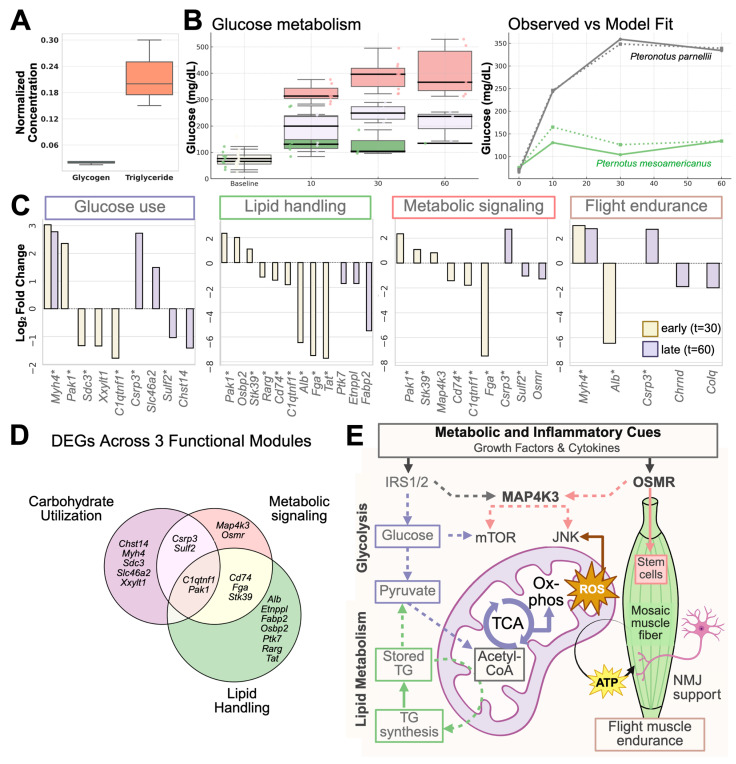
Integrated molecular, physiological, and functional adaptations in flight muscle. (**A**). Normalized concentrations of glycogen and triglycerides measured in *P. parnellii* pectoralis muscle, showing preferential storage of triglycerides. (**B**). Glucose tolerance test (GTT) time series and linear mixed model fit. Left: Blood glucose levels measured at baseline, 10, 30, and 60 min after oral glucose administration across individual bats (*n* = 36). *P. mesoamericanus* (green) shows a uniformly low response, whereas *P. parnellii* exhibits both high (red) and low (gray) glucose clearance phenotypes. Right: Linear mixed-model fit to the GTT time series. Solid lines represent observed species means; dotted lines indicate the model-predicted trajectories. The model shows a significant species × time interaction (*p* < 0.001) in glucose clearance for *P. mesoamericanus* (green, *n* = 10) and *P. parnellii* (dark gray, *n* = 26). (**C**). Log_2_ fold change in gene expression for differentially expressed genes (DEGs) in flight muscle at 30 min (yellow) or 60 min (purple) post-glucose administration compared to baseline (0 min). Only DEGs with |log_2_FC| > 1.5 and adjusted *p*-value < 0.01 (EdgeR) are shown. Genes are grouped into four functional modules: Glucose use, lipid handling, metabolic signaling, and flight endurance based on enrichment clustering using g:Profiler-enriched terms. Genes with (*) are shared in different modules. Module-specific DEGs include *Chst14*, *Slc46a2*, *Xxylt1* (glucose handling), and *Osbp2*, *Ptk7*, *Etnppl,* and *Fabp2* (lipid handling), highlighting metabolic specialization in bat flight muscle. (**D**). Venn diagram illustrating the overlap of DEGs across three functional modules: Carbohydrate Utilization, lipid handling, and metabolic signaling. Genes in overlapping regions are implicated in multiple pathways. Notably, *Map4k3* and *Osmr* are uniquely assigned, representing upstream metabolic and cytokine signaling inputs. (**E**). Schematic of core cellular metabolic pathways supporting flight muscle performance. The illustration highlights the integrated metabolic and structural adaptations required for powered flight. The conceptual model incorporates well-established molecules in glucose metabolism, lipid handling, mitochondrial ATP production, and neuromuscular activity, based on general principles of cell biology. Solid arrows represent direct metabolic inputs; dotted arrows indicate indirect regulatory inputs. Growth factors and cytokine signals act through receptors IRS1/2 and OSMR, which activate downstream protein MAP4K3. These signaling pathways converge on mTOR and JNK proteins to regulate cellular metabolic and stress responses. Pyruvate-derived acetyl-CoA is routed either to the TCA cycle or triglyceride (TG) synthesis. Stored TG serves as an energy reservoir for mitochondrial energy production. ATP fuels neuromuscular junction (NMJ) activity and contraction, supporting muscle endurance. In parallel, OSMR contributes to regenerative signaling, consistent with muscle stem cell activation.

**Table 1 cells-14-01190-t001:** Summary of Established Myoblast Cell Lines Across Vertebrate Species. A comparative overview of established myoblast and muscle-derived cell lines from various vertebrate species, including their typical research uses. These cell lines have been applied in diverse fields such as muscle physiology, regenerative biology, disease modeling, metabolic research, aquaculture, and gene therapy development. This table highlights the broad utility of muscle cell models and emphasizes the gap in non-traditional systems such as bats, which possess unique physiological traits including flight, hibernation, longevity, and enhanced regenerative capacity.

Species/Source	Cell Line Name	Research Applications	References
** *Rat* **	L6	Muscle metabolism, insulin signaling, hypertrophy	[4,5]
** *Mouse* **	C2C12	Myogenesis, muscle regeneration, gene expression	[6]
***Human*** (Myotonic Dystrophy)	Myotonic Dystrophy Cell Lines	Myotonic dystrophy disease modeling, therapy screening	[7,8]
***Dog*** (Myok9)	Myok9	Canine muscle disease and gene therapy studies	[9]
***Dog*** (Dystrophic)	Dystrophic Myoblast Cell Lines	Muscular dystrophy modeling, therapy testing	[10]
***Human*** (Healthy Muscle)	Human Myoblast Cell Line	Muscle aging, physiology, regenerative medicine	[11]
** *Grass Carp* **	CIM	Aquatic muscle growth, stress physiology	[12]
** *Crab-eating* ** ** *macaque* **	NHP iPAX7	Myotube differentiation and muscle regeneration	[13]
** *Japanese Eel* **	JEM1129	Fish muscle development, aquaculture research	[14]
** *Brown-Marbled Grouper* **	EfMS	Fish muscle stem cell and regenerative studies	[15]
** *Chicken* **	chTERT-Myoblasts/Primary Chicken Myoblasts	Avian muscle biology, myogenesis, gene studies	[16]
** *Cuvier’s* ** ** *beaked whale* **	pSV3neo myoblast cell line	Extreme hypoxia, fasting, and deep diving	[17]
** *Bat* **	iBatM-Pmeso-S1 iBatM-Pmeso-TC1	Flight, muscle endurance, muscle stem cell and regeneration, metabolic resilience	This paper

**Table 2 cells-14-01190-t002:** Myoblast growth media.

Media	Base Medium	FBS	Additives	Antibiotics/Antifungals
**GM1**	Ham’s F-10 Nutrient Mix (Gibco (Grand Island, NY, USA), Cat# 11550043)	20% HyClone Characterized FBS (Cytiva (Marlborough, MA, USA), Cat# SH30071.03)	5 ng/mL recombinant human epidermal growth factor (rh EGF) (ATCC (Manassas, VA, USA), Cat #: PCS-999-018), 10 µM dexamethasone (ATCC, Cat #: PCS-999-069), 25 µg/mL recombinant human insulin (ATCC, Cat #: PCS-999-068), 5 ng/mL recombinant human fibroblast growth factor basic protein (rh FGF-b) (ATCC, Cat #: PCS-999-020)	2× Pen/Strep (Gibco, Cat# 15140122), 2 μg/mL Amphotericin B (R&D (Minneapolis, MN, USA), Cat# B23192)
**GM2**	50% Ham’s F-10 Nutrient Mix + 50% DMEM (ATCC, Cat# 30-2002)	20% HyClone Characterized FBS	5 ng/mL rh EGF, 10 µM dexamethasone, 25 µg/mL rh insulin, 5 ng/mL rh FGF-b	2× Pen/Strep, 2 μg/mL Amphotericin B
**GM3**	DMEM	20% HyClone Characterized FBS	5 ng/mL rh EGF, 10 µM dexamethasone, 25 µg/mL rh insulin, 5 ng/mL rh FGF-b	2× Pen/Strep, 2 μg/mL Amphotericin B

**Table 3 cells-14-01190-t003:** Myotube differentiation media.

Media	Base Medium	Horse Serum	Additives	Antibiotics/Antifungals
**DM1**	DMEM	2%	—	2× Pen/Strep, 2 μg/mL Amphotericin B
**DM2**	Ham’s F-10 Nutrient Mix	2%	—	2× Pen/Strep, 2 μg/mL Amphotericin B
**DM3**	DMEM	2%	5 ng/mL rh EGF, 10 µM dexamethasone, 25 µg/mL rh insulin, 5 ng/mL rh FGF-b	2× Pen/Strep, 2 μg/mL Amphotericin B
**DM4**	Ham’s F-10 Nutrient Mix	2%	5 ng/mL rh EGF, 10 µM dexamethasone, 25 µg/mL rh insulin, 5 ng/mL rh FGF-b	22× Pen/Strep, 2 μg/mL Amphotericin B

**Table 4 cells-14-01190-t004:** Comparative features of commonly used skeletal muscle cell models.

Cell Line	Doubling Time	Differentiation Timeline	Advantages/Disadvantages	References
**C2C12 (mouse)**	~18–24 h	Myotubes by day 2–3; Contraction with induction after day 5–6	Differentiation declines with passage; murine model; moderate NMJ relevance	[6,89]
**L6** **(rat)**	~22–30 h	Myotubes by day 5–6; Contraction by day 6 with induction	Reduced sarcomeric structure; lower nAChR expression; NMJ modeling limited	[4,5]
**Primary Myoblasts**	Variable (~24–36 h)	Myotubes by day 8–10; Contraction after day 10 with induction	Short culture lifespan; labor-intensive isolation; slower differentiation	[90]
**iPSCs**	Variable (~36–72 h)	Myotubes > 10 days with induction; variable contraction	Long, heterogeneous differentiation; genomic variability	[91,92]
**iBatM-S1 (bat)**	26.33 h (P8), 28.96 h (P40)	Myotubes by day 2; Spontaneous contraction after day 2	New line; long lifespan, no decline in function with passage; high NMJ relevance	This study
**iBatM-TC (bat)**	23.42 h (P8), 25.46 h (P40)	Myotubes by day 2; Spontaneous contraction after day 2	As above, immortalization effects under evaluation; high NMJ relevance	This study

## Data Availability

Original data underlying this manuscript can be accessed on 25 July 2025 from the Stowers Original Data Repository at https://www.stowers.org/research/publications/LIBPB-2563.

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
