# Peer review of "From Development to Regeneration: Insights into Flight Muscle Adaptations from Bat Muscle Cell Lines"

_cells, 2025, doi:10.3390/cells14151190_

Round 1

Reviewer 1 Report

Comments and Suggestions for Authors

This scientific article ‘From Development to Regeneration: Insights into Flight Muscle Adaptations from Bat Muscle Cell Lines’ established the first two myoblast cell lines from the pectoralis muscle of Pteronotus mesoamericanus bat. These two bat myoblasts lines remain genetically stable over time, Retain the Proliferation and Differentiation Capacity of Bat Primary Myoblasts. Furthermore, The integration of in vivo transcriptomics, functional metabolic data, and in vitro cellular models creates a strong systems-level platform.

Overall, this is a well-written and interesting paper. Their study provides the first in vitro platform for investigating bat muscle research, enabling direct exploration of muscle regeneration, metabolic resilience, and evolutionary physiology. However, there are some comments and recommendations that need to be addressed before this paper is accepted for publication in Cells.

Major comments and recommendations

  1. There is potential confusion between P. mesoamericanus (used for myoblast cell lines development) and P. parnellii (used for transcriptomics and biochemical assays). Add a brief summary table or paragraph in the methods or introduction that clarifies what data came from which species and justify using P. parnellii as a proxy when necessary.
  2. The spontaneous contraction is highlighted but lacks quantification or electrophysiological analysis. Consider adding basic metrics (onset, frequency, amplitude) or acknowledge limitations and future plans in the discussion.
  3. Viral (Genomic Integration of hTERT/CDK4) integration is a common concern. Were integration sites verified or are there any risks of transformation? Briefly discuss this or state it as a limitation for future work.
  4. Some genes are written inconsistently (e.g., Mymx, Etf8 vs MYMX, ETV5).
  5. Video Legends: Include more detailed legends or frame count references in the Videos 1–3 to help readers interpret contraction timing.
  6. Some sections jump from cell work to transcriptomics without clear transitions. Add brief introductory statements linking results would be helpful. For example: “To contextualize these gene expression changes, we next established a complementary in vitro model…”
  7. In the paper, the authors highlight many strengths, but a discussion of limitations would improve transparency. For example: use of P. parnellii instead of P. mesoamericanus for transcriptomics, Absence of direct metabolic flux measurements and so on.
  8. The authors claimed that both lines have been maintained for over 50 passages across 5 months in culture. But did not show the morphology and differentiation capability.  Adding these data would strengthen the cell lines stability.

Other minor comments:

  1. “cylcin-dependent kinase” → “cyclin-dependent kinase”
  2. “compoistion” → “composition”.
  3. “neuromuscular activity as reflected in...” could be rephrased more clearly.
  4. Please double check the spelling, gaps between words, such as ‘Line 98: Together, co-expression ‘of  TERT’ and’
  5. Line 106, ‘Pteronotus mesoamericanus,
  6. when the full name appear the first time in the paper, a short name which would be used later on should be addressed.

Reviewer 2 Report

Comments and Suggestions for Authors

In this manuscript titled “From Development to Regeneration: Insights into Flight Muscle Adaptations from Bat Muscle Cell Lines” by Deng et al., the authors perform RNA-Seq of bat flight muscle tissue samples and derive two immortalized cell lines from bats to establish a novel model to study multiple aspects of muscle physiology. The RNA-Seq data identify genes differentially regulated after glucose challenge, providing insight into bat muscle physiology. The authors characterize the cell lines, demonstrating they are able upon induction to differentiate and providing evidence of contractility. The lines have a stable number of chromosomes and can be maintained without change in cellular structure or differentiation capacity for >40 passages. The data are useful for the muscle field, and establish an important new model for comparative study of muscle physiology. The cell model is further important for understanding differences in vertebrate muscle physiology and will be useful for future studies. The work is of interest to a broad array of biologists working with muscle biology, evolution, and metabolism. I have the following minor suggestions to improve the manuscript.

Line 219: The Supplemental Methods are not available at the provided link, as it directs to a 404: Page Not Found error.  Likewise, I was not able to review the Videos (1-3) of spontaneously contracting myotubes.

In Figure 3, the F-actin staining the the primary cells as P9 looks very regular and sarcomeric, and in the iBatM-Pmeso-TC cells, there is somewhat of a regular pattern, but in the iBatM-Pmeso-S cells, F-actin is in bright, puncta-like structures and does not appear regular and repeating. Can the authors comment or provide better images of differentiated cytoskeletal structures in the P-mesoS and Pmeso-TC cells? The MF20 and Desmin do indicate these are muscle cells, as does the ability to undergo fusion. The DAPI label in E is on top of the image.

The authors show the new bat myoblast lines are able to differentiate and evaluate how serum and media influence differentiation. Is it possible to stain markers to show how far these cells are able to differentiate? C2C12 cells, for example, progress partway through the differentiation program (for example expression MyoD and MyoG and markers of mature muscle), but they do not fully complete the later steps overserved in vivo. Do these bat cells make it further in the differentiation process, and do the authors see expression of presumably conserved markers of muscle differentiation?

Green in Figure 6B refers to P. mesoamericanus, but it is not clear what the red and grey correspond to. If these are the “both high and low glucose clearance phenotypes” observed for P. parnellii, how were these distinguished? What is the justification for plotting as two separate populations, but showing only a single model in the right panel of B?

Line 755: A major point listed in the conclusion is the biphasic response to metabolic overload in muscle. This point warrants additional discussion in the Discussion and integration with the broader literature. Is this biphasic response specific to bat, for example? This response seems to align with findings in other models suggesting a tightly regulated temporal response to glucose, with rapid, intermediate, and longer transcriptional and metabolic responses to glucose stimulation.

The authors present the first example of bat myoblast cell lines, which is very exciting, but the authors should mention bat cell lines from other cell types do exist and support metabolic specialization in bat (for example Bai et al, FEBS Open Bio, 2024; Dejosez, Cell, 2023; and papers already cited such as Jagannathan et al., Elife, 2024 and Alcock et al, Sci Rep, 2024).

Figure S1 – Can the authors show the entire volcano plot? It seems unusual to cut-off the plot at a -log10p-val of 2, obscuring the shape of the rest of the volcano plot and presumably many additional datapoints.

There are small grammar errors throughout the manuscript. A few examples include Line 541, spelling: compoistion should be composition; Line 565, primer should be primary; Line 589 P. mesoamericanus should be in italics.
